# LINE-1 retrotransposons facilitate horizontal gene transfer into poxviruses

M Julhasur Rahman[1], Sherry L Haller[2], Ana MM Stoian[1], Jie Li[3], Greg Brennan[1], Stefan Rothenburg[1]*

[1]Department of Medial Microbiology and Immunology, School of Medicine, University of California, Davis, Davis, United States; [2]Center for Biodefense and Emerging Infectious Diseases, University of Texas Medical Branch, Galveston, United States; [3]Genome Center, University of California, Davis, Davis, United States

**Abstract** There is ample phylogenetic evidence that many critical virus functions, like immune evasion, evolved by the acquisition of genes from their hosts through horizontal gene transfer (HGT). However, the lack of an experimental system has prevented a mechanistic understanding of this process. We developed a model to elucidate the mechanisms of HGT into vaccinia virus, the prototypic poxvirus. All identified gene capture events showed signatures of long interspersed nuclear element-1 (LINE-1)-mediated retrotransposition, including spliced-out introns, polyadenylated tails, and target site duplications. In one case, the acquired gene integrated together with a polyadenylated host U2 small nuclear RNA. Integrations occurred across the genome, in some cases knocking out essential viral genes. These essential gene knockouts were rescued through a process of complementation by the parent virus followed by nonhomologous recombination during serial passaging to generate a single, replication-competent virus. This work links multiple evolutionary mechanisms into one adaptive cascade and identifies host retrotransposons as major drivers for virus evolution.

*For correspondence:
rothenburg@ucdavis.edu

Competing interest: The authors declare that no competing interests exist.

## Editor's evaluation

This landmark paper reports real-time gene acquisition by vaccinia virus, a DNA virus that replicates in the cytoplasm of infected host, from the host DNA genome. The compelling evidence comes from the rescue of a defective vaccinia virus with a cell line that provides an essential function. Then, horizontal gene transfer that bears sequence hallmarks of LINE-1 transposition and subsequent recombination with sibling genomes are required to generate viable genomes. Detection and description of these combinations of rare events is a technical feat that will be of great interest to anyone interested in human or viral evolution.

## Introduction

Horizontal gene transfer (HGT) is the transmission of genetic material between different organisms. This process shapes the evolution and functional diversity of all major life forms, including viruses (*Keeling, 2009*; *Gilbert and Cordaux, 2017*). The genomes of large DNA viruses, including poxviruses and herpesviruses, contain numerous genes that likely have been acquired from their hosts by HGT. Many of these captured genes are involved in immune evasion or protection from environmental damage (*Hughes and Friedman, 2005*; *Bratke and McLysaght, 2008*; *Bratke et al., 2013*; *Wilson and Brooks, 2011*; *Iyer et al., 2006*). So far, the detection of HGT in large DNA viruses has relied on bioinformatic approaches, primarily phylogenetic and sequence composition analyses. For example, Odom et al. performed a family wide comparison of poxvirus coding genes to different

taxonomic subsets, such as 'all eukaryotic genes' or 'all other viral genes'. They found that poxvirus ORFs are on average more similar to eukaryotic genes than other viruses, suggesting substantial HGT into poxviruses over evolutionary time (*Odom et al., 2009*). In an orthogonal approach, Monier et al. used a Bayesian methodology to identify large DNA virus genes with anomalous nucleotide composition relative to the viral genome as a whole. Using this approach, they determined that a large number of the compositionally anomalous genes in *Poxviridae* are associated with host immune control (*Monier et al., 2007*). One of the most striking examples for HGT are viral homologs of host interleukin 10 (IL-10) genes. Viral IL-10 genes, which presumably provide a selective advantage through inhibition and modulation of the antiviral response, have been independently acquired by several herpesviruses and poxviruses (*Hughes and Friedman, 2005*; *Schönrich et al., 2017*). While bioinformatic methods can detect putative HGT they can do little to elucidate the frequency or mechanisms of HGT.

The two most likely mechanisms for HGT into DNA viruses are either a DNA-mediated process such as integration via recombination or the help of DNA transposons, or an RNA-mediated integration such as gene transfer via retrotransposons or retroviruses (*Schönrich et al., 2017*; *Sekiguchi and Shuman, 1997*). While it is unclear how most viral orthologs of host genes were acquired, the vIL-10 gene in ovine herpesvirus 2 has retained the intron structure of the host gene, supporting a mechanism of DNA-mediated HGT in this instance (*Jayawardane et al., 2008*). However, because most herpesvirus genes and all known poxvirus genes lack introns, it is unclear how common DNA-mediated HGT events may be. In fact, the taterapox virus genome contains a host-derived short interspersed nuclear element (SINE), which is flanked by a 16-bp target site duplication (TSD), indicating that the long interspersed nuclear element-1 (LINE-1, L1) group of retrotransposons can also facilitate the transfer of host genetic material into poxviruses through an RNA-mediated process (*Piskurek and Okada, 2007*).

The family *Poxviridae* comprise many viruses of medical and veterinary importance, including variola virus, the causative agent of human smallpox, vaccinia virus (VACV) the best-studied poxvirus, which was used as the vaccine against smallpox, and monkeypox virus, an emerging human pathogen. Poxviruses can enter a large number of mammalian cell types in a species-independent manner; therefore, productive infection largely depends on the successful subversion of the host immune response (*Haller et al., 2014*). More than 40% of poxvirus gene families show evidence that they were host derived, including many genes involved in immune evasion (*Hughes and Friedman, 2005*; *Bratke and McLysaght, 2008*). This high proportion of host-acquired genes suggests that poxviruses may represent ideal candidates to study the process of HGT.

In order to establish an experimental model of HGT, we exploited the dependence of VACV replication on effective inhibition of PKR, which is an antiviral protein activated by double-stranded (ds) RNA formed during infection with both DNA and RNA viruses. Activated PKR phosphorylates the alpha subunit of eukaryotic translation initiation factor 2 (eIF2α), which inhibits cap-dependent protein synthesis, thereby preventing virus replication (*Wu and Kaufman, 1997*). Because PKR exerts a strong antiviral effect, many viruses, including poxviruses, have evolved inhibitors that can block PKR activation (*Langland et al., 2006*). Most poxviruses that infect mammals contain two PKR inhibitors (*Bratke et al., 2013*), which are called E3 (encoded by E3L) and K3 (encoded by K3L) in VACV. E3 is a dsRNA-binding protein, which prevents PKR homodimerization and thereby inhibits PKR activation (*Davies et al., 1993*, *Chang et al., 1992*). K3L, which may itself be a horizontally transferred gene, possesses homology to the PKR-binding S1 domain of eIF2α, and acts as a pseudosubstrate inhibitor of PKR (*Beattie et al., 1991*; *Dar and Sicheri, 2002*).

In this work, we have established a model system to study HGT from the host genome into VACV. In all of the HGT events we detected, the captured genes showed multiple signatures that host LINE-1 retrotransposons were involved in the gene capture. In some cases, integration of the horizontally acquired gene occurred in essential virus genes. In those cases, complementation of both the parental virus and the virus that acquired the gene was necessary for virus replication. After only a few rounds of continued passaging, these complementing viruses recombined to generate single replication-competent viruses. Taken together, retrotransposon-mediated HGT followed by complementation and recombination describes an unexpected cascade of evolutionary processes to rapidly integrate host genes into the viral genome.

## Results

### An experimental system to detect host gene capture by vaccinia virus

In order to detect HGT into VACV, we designed an experimental system that uses the selective pressure imposed by PKR on VACV replication. In this model, we use a VACV strain (VC-R2) that lacks both PKR antagonists and can therefore only replicate in PKR-deficient cells or in complementing cells that express viral PKR inhibitors, such as VACV E3 or K3 (*Hand et al., 2015*). We stably transfected PKR-competent RK13 cells with E3L-tagged mCherry (mCherry-E3L) (*Figure 1A*), selected an individual clone that showed robust mCherry-E3 expression by fluorescence microscopy, and verified that it contained the entire expression cassette (*Figure 1—figure supplement 1*). The PKR-sensitive virus VC-R2 can infect these cells as efficiently as wild-type VACV, but not wild-type RK13 cells. However, if during the course of infection VC-R2-acquired mCherry-E3L from the host cell, the virus would be able to replicate in PKR-competent cells and fluoresce red (*Figure 1A*). The expression cassette also contains the rabbit β-globin intron (*Figure 1B*) enabling us to distinguish between direct DNA-mediated integration or indirect RNA-mediated integration, the two most likely mechanisms for HGT into DNA viruses. We infected the RK13-mCherry-E3L cell line with VC-R2 and titered the resulting viruses on permissive RK13 + E3 + K3 cells (*Rahman et al., 2013*). Then, we infected wild-type RK13 cells, which are nonpermissive for VC-R2, with these viruses and isolated plaques that expressed both mCherry and EGFP (*Figure 1A*). Overall, we identified 20 recombinant viruses with unique integration sites (*Figure 1D, E*). In total, we screened approximately $7.5 \times 10^8$ virions while optimizing this system. After optimization, we screened $3.63 \times 10^8$ virions derived from a single 48-hr infection of RK13 cells (multiplicity of infection [MOI] = 0.01). We screened this population in RK13 cells (MOI = 0.2) and identified 16 unique HGT isolates. This indicates a detected transfer rate of mCherry-E3L into the VACV genome of approximately 1 in 22.7 million viable virions.

### Horizontally acquired genes show evidence of LINE-1-mediated retrotransposition

To map the integration sites, we plaque-purified each HGT candidate three times and used both Sanger sequencing of amplicons from inverse PCR (*Ochman et al., 1988*) and PacBio-based long-read sequencing (*Eid et al., 2009*; *Figure 1—figure supplements 2 and 3*). The integration sites were distributed throughout the VACV genome, and were found in intergenic spaces (three cases), as well as in nonessential and essential genes (*Figure 1D, E*, *Table 1*). All 20 isolates (HGT1–20) displayed hallmarks of RNA-dependent, retrotransposon-mediated integrations, including spliced-out introns and long untemplated stretches of poly(A) at the 3′ end (*Esnault et al., 2000*). To determine if mCherry-E3L was present in both sites of the ITR in the C19L/B25R locus in HGT15, PCR with DNA of HGT15 and VC-R2, as control, was performed with a primer located in the ITR, toward the termini of the integration site and primers outside of the ITR. Both primer combinations showed larger bands when DNA HGT15 DNA was used as template, in comparison to VC-R2 DNA, which is indicative of mCherry-E3L in both ITR copies (*Figure 1—figure supplement 4*). In HGT20, the integrated sequence contained the intron-less mCherry-E3L cassette fused to an 88-bp cellular DNA fragment encoding U2 small nuclear (sn) RNA, which contained a second poly(A) tract (*Figure 1E*, *Figure 1—figure supplement 5*).

In 19 cases, short TSDs of VACV sequences surrounded the integration sites (*Figure 1C, E*, *Figure 2*, *Supplementary file 1*). These TSDs had an average length of 15.9 bp and a consensus (3′-AA/TTTT-5′) motif at the integration site typical of the cleavage consensus site of the LINE-1 (L1) group of retrotransposons (*Figure 2*; *Kojima, 2010*; *Gilbert et al., 2002*). HGT11 contained a 2.7-kb deletion immediately downstream of the integration site in B4R, precluding the identification of a TSD (*Figure 1E*).

### HGT acquisition in essential genes is facilitated by mutual complementation

To analyze whether HGT imposed any fitness costs, we compared the plaque sizes of HGT viruses in RK13 and BSC-40 cells with plaques formed by replication-competent viruses VC-2 and vP872 (*Beattie et al., 1991*). For most HGT viruses, plaque sizes were comparable to the replication-competent viruses in RK13 cells (*Figure 3A*). However, in BSC-40 cells some viruses formed substantially smaller plaques than in RK13 cells, for example HGT3, HGT14, and HGT19. HGT13 (insertion in F13L) formed

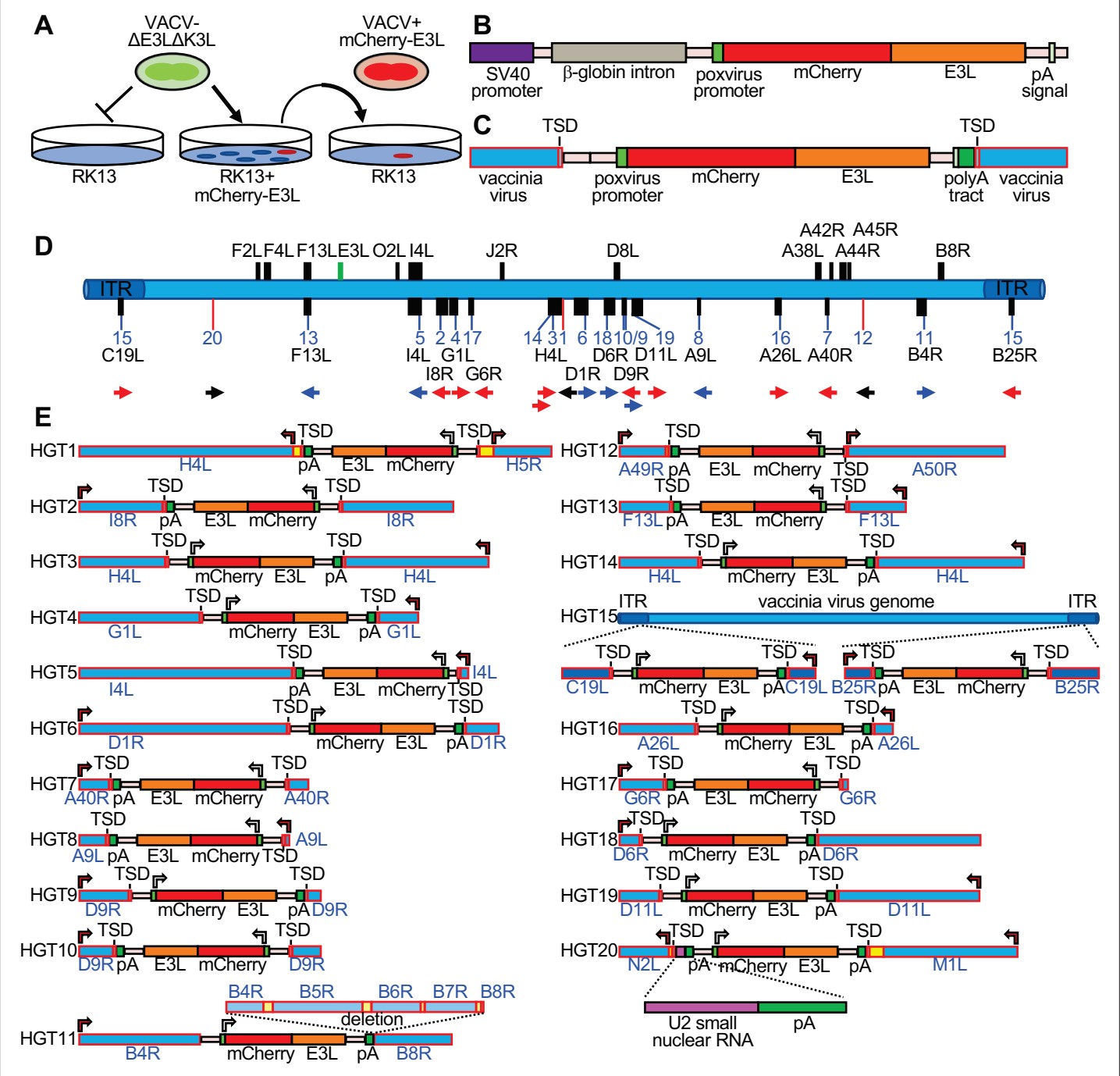

**Figure 1.** Detection of experimental horizontal gene transfer (HGT) in vaccinia virus. (**A**) VACV lacking E3L and K3L cannot replicate in wild-type RK13 cells, but can replicate in cells stably transfected with mCherry-E3L. Virus that acquired E3L can replicate in wild-type RK13 cells (right). (**B**) Schematic of the mCherry-E3L vector that was stably transfected into RK13 cells. (**C**) Schematic of the general genetic architecture of horizontally transferred genes identified in VACV isolates. (**D**) HGT integration sites in the VACV genome. The VACV genome is represented in blue. Genes highlighted above the genome were described to have likely originated from HGT (*Hughes and Friedman, 2005*; *Bratke and McLysaght, 2008*). Features shown below the genome are: integration sites into genes (black boxes, blue lines) or into intergenic regions (red lines). The orientation of the transferred genes is indicated by the arrow, colors of arrows indicate the orientation of mCherry-E3L relative to target genes (blue: same direction; red: opposite direction; black arrow: intergenic). (**E**) Maps of mCherry-E3L integration sites in HGT1–20. Arrows indicate the direction of transcription for VACV and mCherry-E3L. Intergenic regions are depicted in yellow. The position of an integrated U2 small nuclear RNA and the associated poly(A) tract is shown for HGT20 by dashed lines.

The online version of this article includes the following source data and figure supplement(s) for figure 1:

*Figure 1 continued on next page*

*Figure 1 continued*

**Figure supplement 1.** Stable expression of mCherry-E3L in RK13 cells.

**Figure supplement 1—source data 1.** PCR for mCherry-E3L.

**Figure supplement 2.** Sequences of the pmCherry-E3L plasmid, VACV genomic sequences surrounding the integration sites and the integrated genes in HGT1–20.

**Figure supplement 3.** Map of pmCherry-E3L.

**Figure supplement 4.** HGT15 contains one mCherry-E3L copy in each inverted terminal repeat (ITR).

**Figure supplement 4—source data 1.** PCR with primers spanning the mCherry-E3L integration site in HGT15 in comparison to VC-R2.

**Figure supplement 5.** Integration site and composition of the U2-mCherry/E3L fusion in HGT20.

tiny plaques in both cell lines, consistent with a previous report of defective viral spread in a F13L-deficient VACV (*Blasco and Moss, 1991*). HGT11, containing a deletion stretching from B4R through B8R, also exhibited a small plaque phenotype in both cell types. These findings indicate that, unsurprisingly, HGT can cause a loss-of-fitness.

Surprisingly, eight of the VACV genes that were disrupted by HGT were reported to be essential for virus replication (*Table 1*).

Because these insertions should presumably inactivate these essential genes, it was unclear how this HGT resulted in viable viruses. One possibility could be that although these viruses were plaque purified three times, the parent virus may have persisted to complement the otherwise nonviable HGT virus. To test this hypothesis, we infected RK13 + E3 + K3 cells, which are also permissive for the parental virus, with each plaque-purified HGT isolate. In all these infections, we identified the expected EGFP/mCherry double-positive foci. Additionally, foci that were EGFP positive but mCherry negative

**Table 1.** Integration sites of mCherry-E3L in horizontal gene transfer (HGT) viruses and importance of disrupted genes for virus replication.

| Isolate # | Integration site | Essential vs. nonessential |
|---|---|---|
| HGT1 | Intergenic between H4L and H5R | Presumably nonessential |
| HGT2 | I8R | Essential (*Gross and Shuman, 1996*) |
| HGT3 | H4L | Essential (*Kane and Shuman, 1992*) |
| HGT4 | G1L | Essential (*Hedengren-Olcott et al., 2004*) |
| HGT5 | I4L | Nonessential (*Child et al., 1990*; *Gammon et al., 2010*) |
| HGT6 | D1R | Essential (*Hassett et al., 1997*) |
| HGT7 | A40R | Nonessential (*Wilcock et al., 1999*) |
| HGT8 | A9L | Essential (*Yeh et al., 2000*) |
| HGT9 and HGT10 | D9R | Nonessential (*Parrish and Moss, 2006*) |
| HGT11 | B4R | Nonessential (*Burles et al., 2014*) |
| HGT12 | Intergenic between A49R and A50R | Presumably nonessential |
| HGT13 | F13L | Nonessential (*Blasco and Moss, 1991*) |
| HGT14 | H4L | Essential (*Kane and Shuman, 1992*) |
| HGT15 | C19L/B25R | Nonessential (*Perkus et al., 1991*) |
| HGT16 | A26L | Nonessential (*Howard et al., 2008*; *Chang et al., 2019*) |
| HGT17 | G6R | Nonessential (*Senkevich et al., 2008*) |
| HGT18 | D6R | Essential (*Broyles and Fesler, 1990*) |
| HGT19 | D11L | Essential (*Seto et al., 1987*) |
| HGT20 | Intergenic between N2L and M1L | Presumably nonessential |

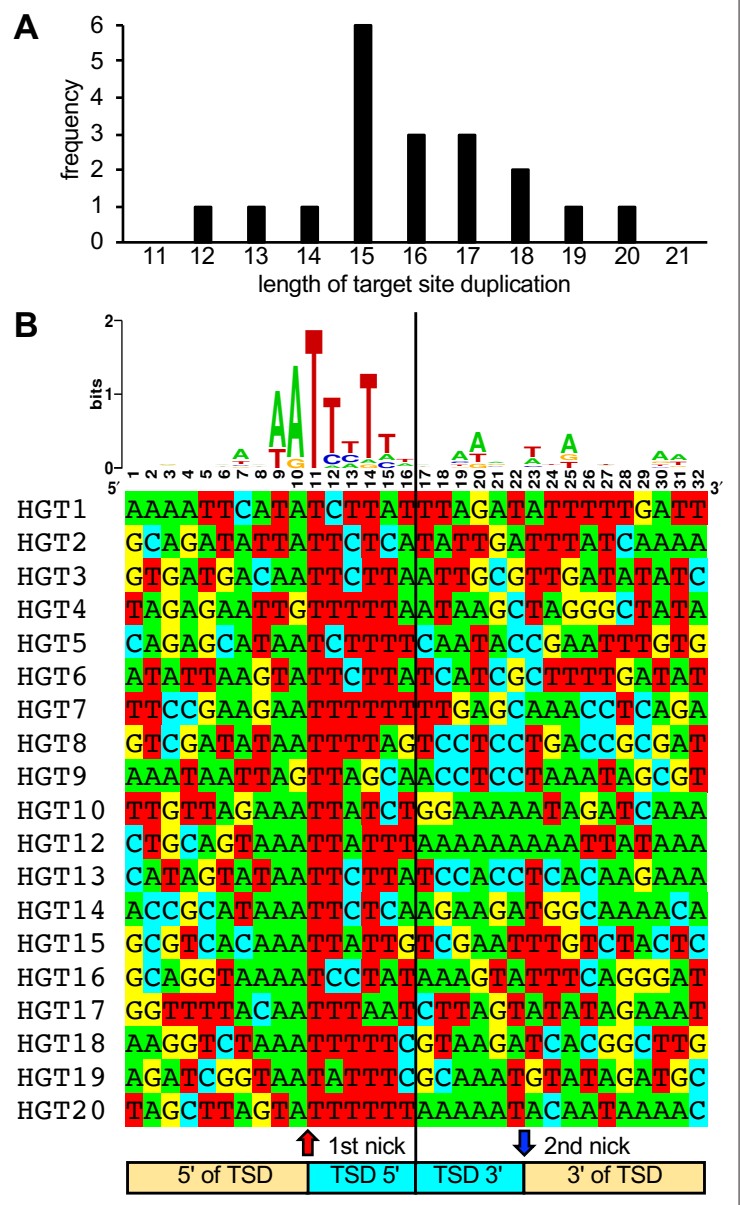

**Figure 2.** Length distribution and composition of target site duplications (TSDs) surrounding integration sites. (**A**) Length distribution of TSDs. (**B**) WebLogo of the consensus sequence and individual sequences of TSDs. Ten nucleotides 5' and 3' of the TSDs, as well as six nucleotides of the TSDs adjacent to the putative first and second long interspersed nuclear element-1 (LINE-1) endonuclease cleavage sites, respectively, are shown.

were observed in isolates in which mCherry-E3L insertions occurred in essential genes, indicating the continued presence of parental virus. The percentage of mCherry positive to mCherry negative foci in these viruses ranged between 7% and 51% (*Figure 3B*, *Figure 3—figure supplement 1*). We hypothesized that the two fluorescent phenotypes indicated mutual complementation, wherein the HGT virus supplied mCherry-E3 and VC-R2 supplied the disrupted essential gene product, thus permitting replication of both viruses in coinfected cells. To test this hypothesis, we coinfected RK13 cells with VC-R2 and the F13L-disrupted HGT13 virus, which did not contain detectable VC-R2 but formed only minute foci in either RK13 or BSC-40 cells. As predicted, in the coinfection we observed larger foci in addition to the very small foci produced by HGT13 alone (*Figure 3C*). In an additional, PCR-based test of this hypothesis, we amplified DNA spanning the integration sites in HGT1, which acquired mCherry-E3L in an intergenic region, or HGT3, in which the essential gene H4L was disrupted. As expected, HGT1 yielded a single 2.3 kB amplicon of the expected size for mCherry-E3L integration. However, we

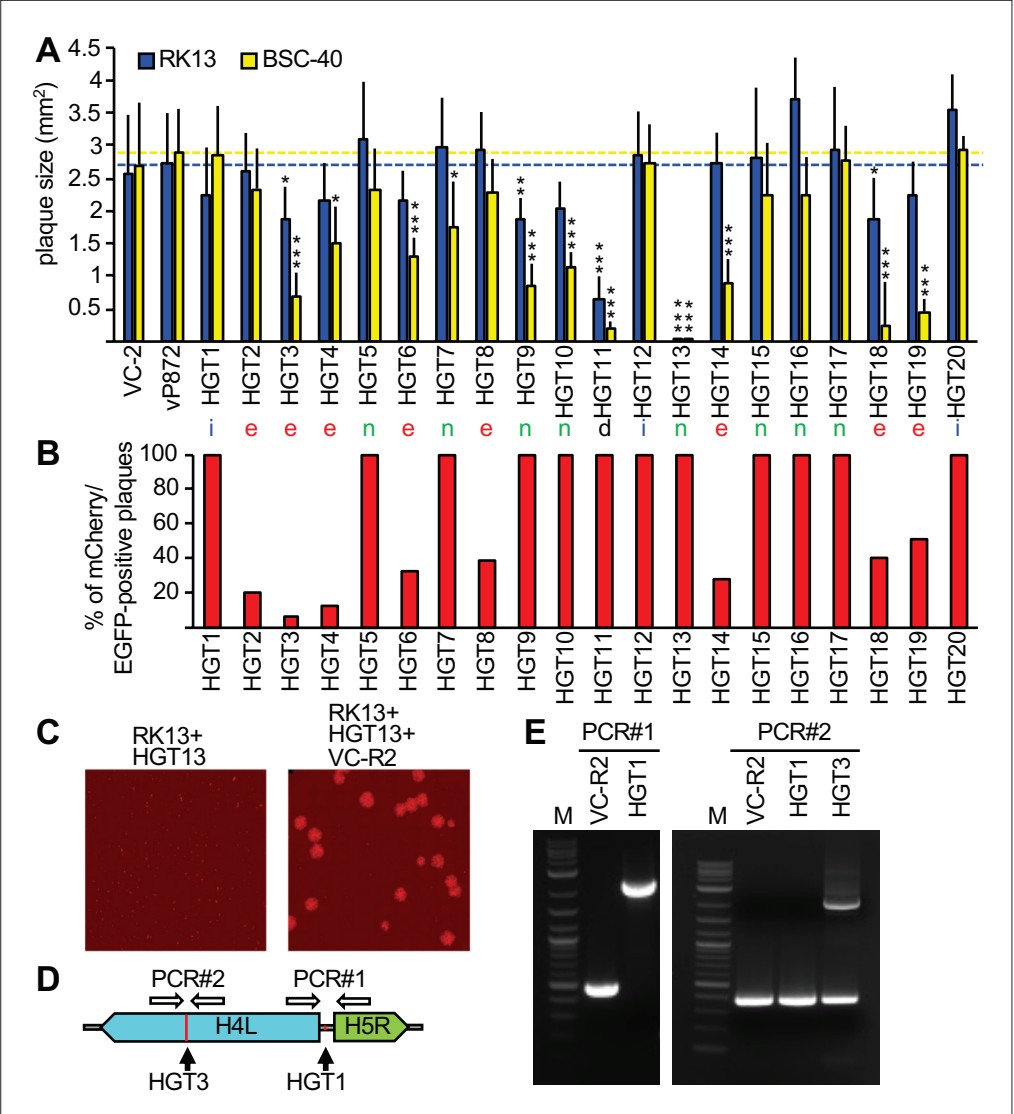

**Figure 3.** Complementation of viruses after mCherry-E3L integration into essential genes. (**A**) Plaque sizes formed by horizontal gene transfer (HGT) virus isolates in RK13 and BSC-40 cells 3 days after infection. Location of integration (intergenic spaces [i], essential [e], nonessential [n], and a deletion [d]) are indicated below each bar. Dotted lines indicate plaque sizes caused by vP872. Between 30 and 387 plaques were measured for each virus (see **Figure 3—source data 1** for details). Mean standard deviations and adjusted p-values calculated with Dunn's test of multiple comparisons are indicated. Asterisks denote significant differences between vP872 and HGT isolates (*p < 0.05; **p < 0.005; ***p < 0.0005). (**B**) Percentage of foci that were mCherry-positive 3 days after infecting RK13 + E3 + K3 cells with the indicated HGT viruses. (**C**) Complementation of HGT13 by VC-R2. RK13 cells were infected with either HGT13 alone (left) or coinfected with HGT13 and VC-R2 (right). Foci were visualized 3 days after infection at the same magnification. (**D**) Location of PCR primers for amplifying integration sites in HGT1 and HGT3. (**E**) PCR amplification for integration site identified for HGT1 (PCR#1, left) and HGT3 (PCR#1, right).

The online version of this article includes the following source data and figure supplement(s) for figure 3:

**Source data 1.** Plaque size data from RK13 and BSC-40 cells.

**Source data 2.** p values of plaque size differences between vP872 and horizontal gene transfer (HGT) isolates in RK13 cells.

**Source data 3.** p values of plaque size differences between vP872 and horizontal gene transfer (HGT) isolates in BSC-40 cells.

**Source data 4.** Number of plaques expressing either mCherry and EGFP, or only EGFP.

*Figure 3 continued on next page*

*Figure 3 continued*

**Source data 5.** PCR amplification for integration site identified for HGT1 (PCR#1, left) and HGT3 (PCR#1, right).

**Figure supplement 1.** mCherry- and/or EGFP-positive foci formed by HGT1–20 on RK13 + E3 + K3 cells.

amplified two products from HGT3: a 396-bp amplicon, consistent with wild-type H4L, and a 2.2 kb amplicon consistent with mCherry-E3L integration into H4L (*Figure 3D, E*). Taken together, these assays support our hypothesis of mutual complementation of these viruses.

## Recombination of complementing viruses generates replication-competent individual viruses

If both viruses complemented each other efficiently, the expected ratio of HGT virus to parent to virus should be close to 1:1. However, for most HGT isolates this ratio was lower, indicating that virus complementation did not necessarily result in optimal replication of both viruses. In these cases, there may be strong selective pressure to evolve a single replication-competent virus that is not reliant on a complementing virus. To test this hypothesis, we serially passaged the HGT3/parental virus population in PKR-competent RK13 cells. In three independent replicates, we observed a rapid increase in double-positive fluorescent foci (*Figure 4A, B*), consistent with an increase in HGT3 fitness. In order to directly test for a fitness increase, we infected RK13 and BSC-40 cells with the passaged virus populations for 48 hr and then titered the viruses on RK13 + E3 + K3 cells. Whereas in RK13 cells the passaged viruses showed only a modest increase in replication (about fivefold) in passages 9 and 17, an approximately 100-fold higher replication was observed for BSC-40 cells infected with passages 9 and 17 as compared to passage 0 (*Figure 4C and D*).

To investigate possible structural changes around the disrupted H4L locus, we designed outward-facing primers in H4L similar to an approach to detect gene amplification in poxviruses (*Elde et al., 2012*; *Brennan et al., 2014*). This PCR strategy only yields products if two or more copies of H4L are adjacent to each other (*Figure 5A*). We amplified products from passaged virus DNA, but not from DNA isolated from VC-R2 or the original plaque-purified virus (P0) (*Figure 5B*). Because of this evidence for genomic structural changes, we used PacBio long-read sequencing to more completely define these changes in the different populations. This approach demonstrated that in each HGT3 virus population, there were genomes in which an intact copy of H4L existed in close proximity to the mCherry-E3L-disrupted H4L locus, although we found several different recombinant viruses with different genomic architectures (*Figure 5C*). Only three shared nucleotides are present at the breakpoints, indicating nonhomologous recombination between the parent and HGT viruses (*Figure 5—figure supplement 1*). Taken together, these observations suggest that recombination occurred early in the population during serial passage, supporting the hypothesis that there may be selective pressure to generate a single viable virus. Alternatively, mCherry-E3L may have integrated into a preexisting H4L duplication; however, this scenario is unlikely, because virus complementation would be unnecessary.

## Signatures of naturally occurring LINE-1-mediated HGT in the Golgi anti-apoptotic protein encoding gene

To determine whether this mechanism is relevant in nature, we asked whether we could detect evidence of RNA-mediated HGT in any phylogenetically predicted HGT candidate genes. We selected and analyzed the phylogenetically predicted HGT gene encoding Golgi anti-apoptotic protein (GAAP, also known as Transmembrane BAX Inhibitor Motif Containing 4, TMBIM4) in the cowpox virus Finland_2000_MAN strain, which is also present in some other orthopoxviruses. GAAP shows approximately 76% protein identity with its mammalian homologs and therefore was likely acquired relatively recently (*Gubser et al., 2007*). Analyzing this gene for signatures of HGT, we identified putative remnants of a poly(A) tail (16 out of 21 bp) 3′ of the gene, and a perfect 18-bp TSD flanking the gene, which contains AA/TTGT at the presumed site for the first nick during integration similar to the AA/TTTT motif identified in our screen. Moreover, adjacent to each TSD are two 59-bp homology regions, which share 83% nucleotide identity, and could be remnants of a larger duplication due to recombination, as seen in this study (*Figure 6*).

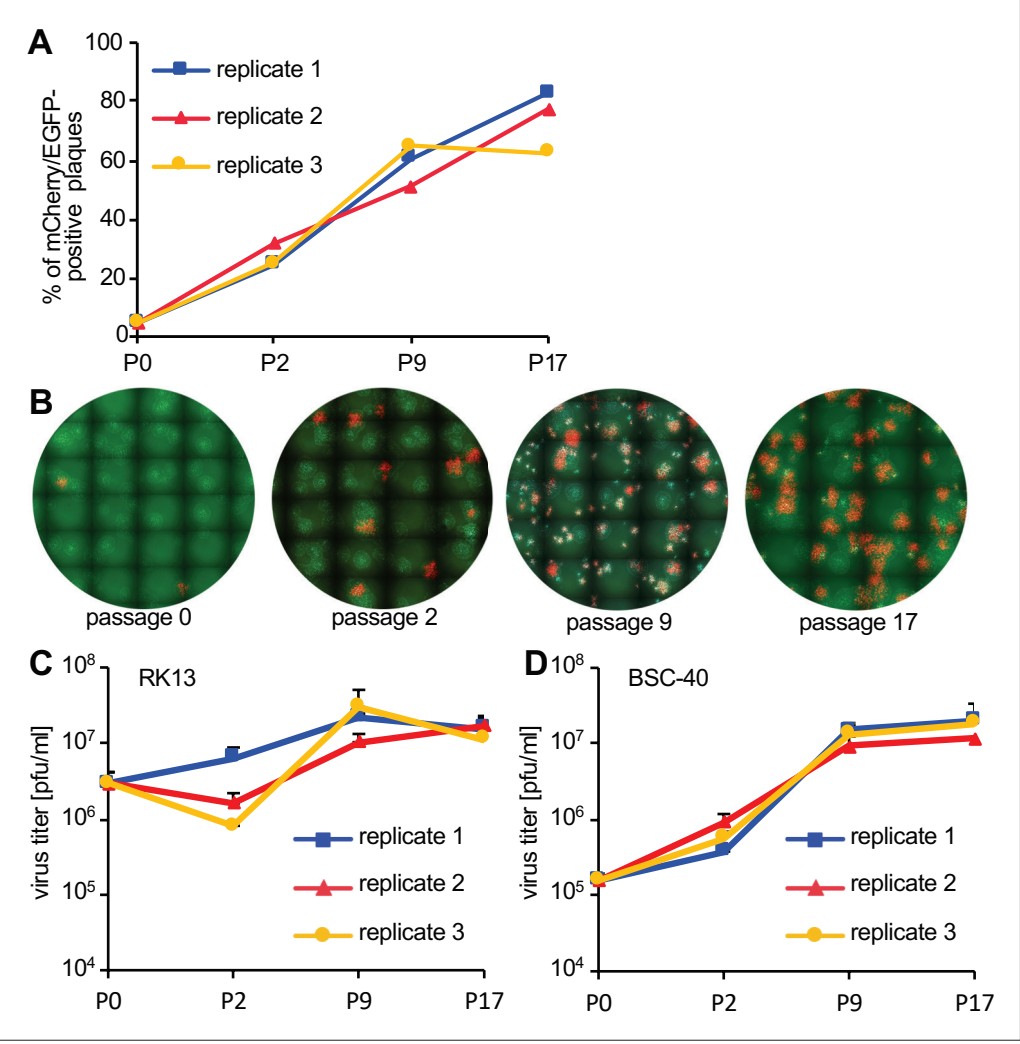

**Figure 4.** Fitness increase of HGT3 after serial passaging. (**A**) The percentage of mCherry-positive foci formed on RK13 + E3 + K3 cells by three serially passaged replicates of HGT3. Percentages were determined for the indicated passages (P). (**B**) Increase in double-positive mCherry and EGFP expressing foci of replicate 3 during serial passaging. (**C**) RK13 and (**D**) BSC-40 cells were infected with serially passaged HGT3 in duplicate (multiplicity of infection [MOI] = 0.01) for 48 hr and viruses were titered on RK13 + E3 + K3 cells. Error bars indicate standard deviations.

The online version of this article includes the following source data for figure 4:

**Source data 1.** Number of plaques expressing either mCherry and EGFP, or only EGFP after serial passaging.

**Source data 2.** Titers of passaged virus populations in RK13 cells.

**Source data 3.** Titers of passaged virus populations in BSC-40 cells.

## Discussion

In this study, we show that HGT from the host cell into poxviruses can be recapitulated in an experimental system. All HGT events showed evidence for an RNA-mediated mechanism, likely mediated by LINE-1 retrotransposons, as they contain TSD of approximately 16 bp surrounding the integration sites, as well as spliced-out introns and untemplated poly(A) tracts 3' of the integrated genes. LINE-1s are ancient and ubiquitous retrotransposons, which are found in both plants and animals. The number of active LINE-1s varies across species (*Ivancevic et al., 2016*). Active LINE-1s encode two proteins called ORF1p and ORF2p, which possess either RNA-binding properties or endonuclease and reverse transcriptase (RT) activity, respectively (*Dombroski et al., 1991*, *Khazina et al., 2011*;

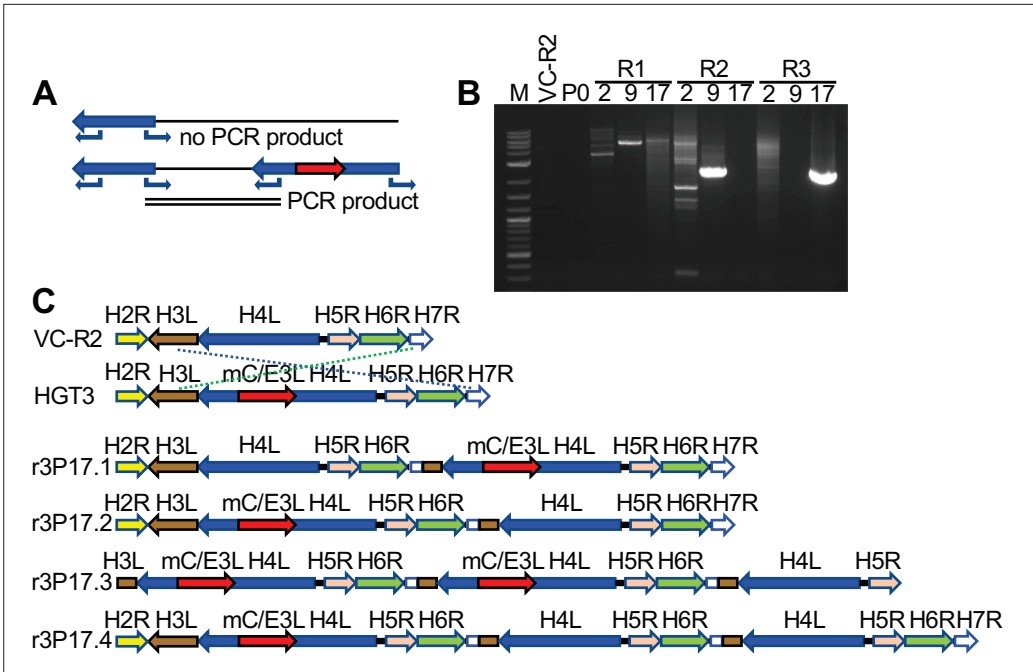

**Figure 5.** Recombination of complementing viruses after serial passaging. (**A**) PCR strategy to detect gene arrays with outward-facing primers. (**B**) Representative PCR products amplified with outward-facing primers for VC-R2 and three independent HGT3 passages. (**C**) Recombination of complementing viruses resulted in tandem arrays of the H4L locus as determined by long-read sequencing.

The online version of this article includes the following source data and figure supplement(s) for figure 5:

**Source data 1.** PCR products amplified with outward-facing primers for VC-R2 and three independent HGT3 passages.

**Figure supplement 1.** Recombination sites in passaged HGT3.

*Doucet et al., 2010*; *Feng et al., 1996*; *Mathias et al., 1991*). LINE-1-mediated reverse transcription is generally thought to occur in the nucleus. Because poxviruses replicate exclusively in the cytoplasm, the detected signatures of LINE-1-mediated retrotransposition after HGT reported here suggest that LINE-1 RT and integrase activities are not limited to the nucleus. The notion that LINE-1s can facilitate the transfer of host genetic material into poxviruses through an RNA-mediated process during the

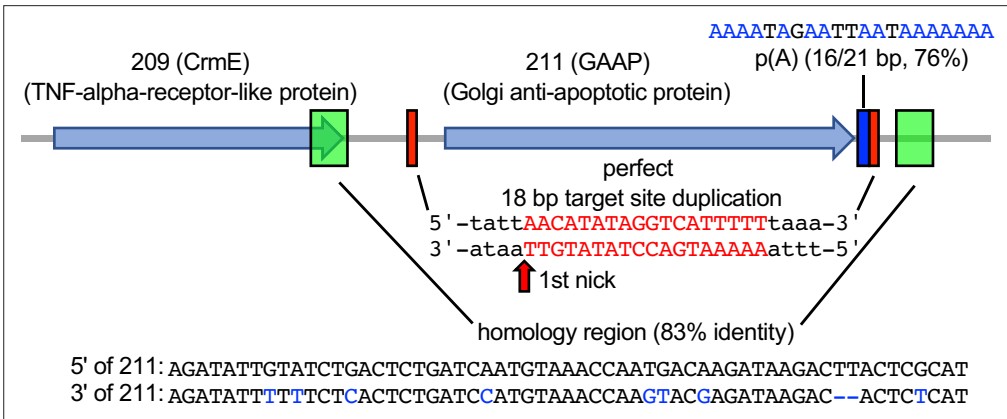

**Figure 6.** Signatures of retrotransposon-mediated horizontal gene transfer (HGT) adjacent to cowpox virus gene 211. The 211 gene encodes Golgi anti-apoptotic protein (GAAP). Red boxes mark a perfect 18-bp putative target site duplication (TSD). p(A) indicates a polyadenylate stretch (16/21 bp) adjacent to the TSD. Green boxes show a duplicated 59 bp stretch with 83% sequence identity (homology region).

natural course of viral infection is further supported by the presence of a naturally occurring SINE element in taterapox virus, which is surrounded by a 16-bp TSD (*Piskurek and Okada, 2007*). This observation shows that poxvirus evolution can be impacted by host transposable elements providing them with selective advantages and accelerating their evolution. It also defines a new function for retrotransposons, in addition to their roles in genomic and epigenetic diversification of the host (*Beck et al., 2011*).

Even though phylogenetically predicted HGT candidates in VACV are distributed throughout the genome (*Figure 1D*; *Hughes and Friedman, 2005*; *Bratke and McLysaght, 2008*), a common notion in the field is that uptake of new genes into poxviruses would be biased toward the ends of poxvirus genomes, because the central genome region contains more essential genes and shorter intergenic spaces and thus insertions there would be less likely to result in viable progeny (*Yao and Evans, 2001*; *Lefkowitz et al., 2006*). Also, poxvirus lineage-specific genes tend to be more often found near the termini of poxvirus genomes (*McLysaght et al., 2003*). However, the integration sites that we discovered in this study were distributed throughout the VACV genome, which indicates that integration per se is not disfavored in the central genomic region. The concentration of insertions around the H4L locus might indicate that some regions of the genome are indeed hotspots for integration. However, the number of integration events detected in this study precludes a definitive answer, and will require further analysis of more HGT events. It is also possible that the insertion site is influenced by the cell type. Because complementation of replication-deficient viruses was more efficient in RK13 as opposed to BSC-40 cells, it appears that propagation of viruses with insertions in essential genes might be favored in the former cells. In a companion paper, Fixsen et al. used a similar strategy to detect HGT by expressing the other VACV PKR inhibitor K3 in RK13 cells, which were also used in our study, and selecting for virus replication in Syrian hamster BHK cells. They found a lower proportion of HGT integrations in the central genome region (*Fixsen et al., 2020*). The reasons for the different integration patterns could be due to the different cells used for selection and might reflect differences in the relative importance of viral genes in these cells or different, cell line-dependent complementation efficiencies.

Our results show that HGT into essential poxvirus genes can result in viable progeny virus in a process that we call cascade evolution. This process sequentially combines HGT, virus complementation and recombination, three classic mechanisms of virus evolution (*Figure 7*). Our experimental system relies on the complete failure for viruses to replicate in the absence of a PKR inhibitor, creating a strong selective pressure. In eight instances in this study we also identified integrations into essential genes, creating a similarly strong selective pressure against virus replication. In these cases, complementation followed by recombination was sufficient to overcome this selective pressure in vitro. The homology regions surrounding the cowpox virus GAAP gene (*Figure 6*) suggest a similar process can occur during natural infection as well. In instances in which the selective pressure to maintain the horizontally transferred gene is weaker or the inactivated gene is nonessential but still contributes to fitness, we hypothesize that similar molecular events might occur. This balance between the negative effects of genome disruption and improved fitness might also display some degree of cell and species specificity.

Our PacBio analyses identified tandem copies of the integrated gene and intact gene, in both orientations relative to each other. However, at the population level the genomic structure surrounding this locus was complex. For example, we identified loci including both intact H4L and the horizontally

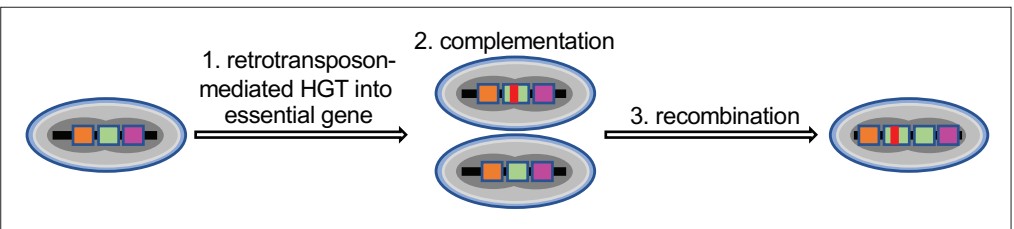

**Figure 7.** Cascade virus evolution after horizontal gene transfer into an essential gene. After retrotransposon-mediated horizontal gene integration into an essential locus (1) the virus initially depends on a helper virus to complement the essential gene (2), which is followed by recombination and results in a single virus with increased fitness.

acquired mCherry-E3L in which multiple copies of each gene were found in cis. Furthermore, in some genomes we identified three or more copies of some genes adjacent to the integration site. In addition to the immediate impact of generating a replication-competent virus through recombination, these variant products of recombination might provide the raw material for subsequent sub- or neofunctionalization of the additional gene copies (*Bayer et al., 2018*).

The majority of the HGT viruses that required complementation with the parental virus formed larger plaques in RK13 cells as compared to BSC-40 cells. This difference may indicate that the complementation is more efficient in RK13 cells. In line with this explanation, we observed that the parent HGT3 population replicated poorly in BSC-40 cells, and the passaged population replicated about 100-fold more efficiently, whereas in RK13 cells, the parent HGT3 population replicated better and only gained a modest increase in replication efficiency during passaging. It is striking that for most observed cases of virus complementation, the ratio of parent to HGT virus was skewed to the parent virus, in some cases making up more than 90% of the population (*Figure 3B*). This observation suggests that there may be competition for cellular resources or viral gene products between the coinfecting viruses. While mCherry-E3 localizes to the cytoplasm and likely antagonizes PKR equally well for each coinfecting virus, the transcription and translation of many other viral gene products are tightly associated with viral factories derived from a single genome (*Katsafanas and Moss, 2007*). Conceptually, this competition may occur when the complementing molecule is required to diffuse or to be transported into another viral factory to exert its effect, or if the viral gene product is not delivered until virus factories fuse late in the replication cycle (*Paszkowski et al., 2016*).

An interesting observation was that in one of the HGT isolates, mCherry-E3L was fused to a polyadenylated fragment of the cellular U2 snRNA. This genetic structure may have been generated by the fusion of mCherry-E3L RNA with U2 snRNA prior to integration. This observation suggests that other cellular genes could piggyback during HGT, which suggests in rare instances multiple genes may be integrated into a single genome. Of note, snRNAs encoding DNA including U1, U2, U5, and U6 can be found in host genomes fused to LINE-1 as a result of RNA ligation (*Buzdin et al., 2003*; *Garcia-Perez et al., 2007*; *Doucet et al., 2015*; *Moldovan et al., 2019*). However, it is unclear what, if any, impact the potential to integrate multiple genes would have on viral evolution.

The calculated approximate transfer rate of E3L-mCherry into VACV in this experimental setting of 1 in 23 million viable virions is probably an underestimation, as integrations in essential loci have detrimental effects on virus replication unless complementing viruses are present. It should be noted that we used a system with a strong synthetic poxvirus promoter and a strong PKR inhibitor in order to detect HGT. Retention of a transferred host gene that has not been optimized is expected to happen with a far lower frequency. However, we selectively looked for the uptake of only one of several thousand genes that are expressed by the host cells. Thus, HGT of host genes, including transposable elements such as SINEs and LINEs, into poxviruses can be expected to occur at a high frequency, but in most cases the inserted gene will not be maintained because of negative or neutral effects on virus replication.

This work, and the companion study by *Fixsen et al., 2020*, provides the first experimental evidence of HGT into poxviruses. The experimental system described here can be extended to study the evolution of transgenes after HGT. In addition, it can be adapted to elucidate the mechanisms of HGT in other groups of viruses for which HGT is believed to play an important evolutionary role. While we only detected RNA-mediated HGT facilitated by LINE-1, it is possible that in other viruses or cell types additional RNA or DNA transposons, or recombination might be important. Similarly, other facilitators, such as endogenous or exogenous retroviruses may also promote HGT and increase virus fitness. Taken together, our work demonstrates that cascade evolution is a powerful and rapid mechanism to integrate new genetic material in poxviral genomes. It also demonstrates that HGT in poxviruses is traceable in experimental settings, and that this mechanism may be relevant to other viruses, which appropriate host genes for their own ends in the ongoing host–virus arms race.

# Materials and methods
## Cell lines
RK13 cells (ATCC CCL-37) and BSC-40 (ATCC CRL-2761) cell lines were kindly provided by Dr. Bernard Moss and Dr. Adam Geballe, respectively. RK13 cells expressing E3 and K3 (designated RK13 + E3

+ K3) were previously described (*Rahman et al., 2013*). All cell lines were maintained in Dulbecco's modified essential medium (DMEM, Life Technologies) supplemented with 5% fetal bovine serum (FBS, Fisher) and 25 µg/ml gentamycin (Quality Biologicals) at 37°C and 5% $CO_2$. RK13 + E3 + K3 cells were supplemented with 500 µg/ml geneticin and 300 µg/ml zeocin (Life Technologies). Cell lines were seeded in 6-well (at $5 \times 10^5$ cells/well) or 12-well (at $2.5 \times 10^5$ cells/well) plates 24 hr before infections. Cell lines used in this study were negative for mycoplasma contamination as determined with Lookout Mycoplasma PCR Detection Kit (Millipore Sigma). During the course of this study, the RK13 cells we used were confirmed to be of European rabbit (*Oryctolagus cuniculus*) origin by PacBio sequencing (ArrayExpress accession: E-MTAB-9682). PKR expressed in BSC-40 was amplified from cDNA and sequenced, which confirmed that the cells are of African green monkey (*Chlorocebus aethiops*) origin.

RK13 cells expressing mCherry-E3 (RK13-mCherry-E3L) were generated by stable transfection of *ApaI*-linearized pmCherry-E3L using GenJet (SignaGen) according to the manufacturer's protocol. The pmCherry-E3L vector contains, in 5′ to 3′ order, the SV40 promoter, rabbit β-globin intron, a synthetic poxvirus early/late promoter, a mCherry-E3L fusion gene, cloned into the backbone of pEGFP-N1 (Clontech). During this process, the CMV promoter and EGFP were removed from pEGFP-N1. Transfected cells were selected with 500 µg/ml geneticin (G418, Invitrogen) for 2 weeks. Individual clones were isolated by seeding cells at a low density (0.3–1 cell/ well) in 96-well plates. Wells containing a single cell, verified by fluorescent microscopy, were selected to establish clonal cell lines. Genomic DNA was prepared to verify the structure of the integrated cassette by PCR with primer JR253-intron-4F (5′-TTC CAG AAG TAG TGA AGA GGC TT-3′) and JR254-E3L-5R (5′-AGC AAG TAA AAC CTC TAC AAA TG-3′). All subsequent experiments were carried out with one RK13 + mCherry-E3L cell clone that showed robust mCherry-E3 expression (*Figure 1—figure supplement 1*).

## Viruses

Vaccinia virus VC-2 (Copenhagen strain) and vP872 (ΔK3L) (*Beattie et al., 1991*) were kindly provided by Dr. Bertram Jacobs and propagated in RK13 cells. VC-R2, which lacks both E3L and K3L (*Brennan et al., 2014*), was propagated in RK13 + E3 + K3 cells. VC-2, vP872, and VC-R2 were purified by zonal sucrose gradient centrifugation as previously described (*Cotter et al., 2017*). Virus clones HGT1 to HGT20 were plaque purified three times in RK13 cells with carboxymethyl cellulose (CMC, 1% final concentration in DMEM + 5% FBS) overlay. Virus clones HGT1–7 were purified by zonal sucrose gradient centrifugation. For HGT8–20, crude cell lysates were used as stocks. All viruses were titered in RK13 + E3 + K3 cells on 12-well plates. Virus samples were tenfold serially diluted and cells were infected in duplicate. One hour after infection, the infecting medium was replaced with CMC and incubated for 3 days at 37°C in 5% $CO_2$. The CMC overlays were then aspirated, and plaques were identified by staining the monolayers with 0.1% crystal violet in 20% methanol.

Serial passaging of HGT3 was performed as described (*Brennan et al., 2014*; *Elde et al., 2012*). Briefly, confluent RK13 cells in 6-well plates were infected with plaque-purified HGT3 virus (MOI = 0.1) in triplicate. Forty-eight hours after infection (hpi), cell lysates were collected, freeze-thawed three times, sonicated, and titered on RK13 + E3 + K3 cells before serially infecting new RK13 cells. Genomic DNA from passaged viruses was isolated from infected RK13 cells (MOI = 1.0) as previously described (*Esposito et al., 1981*). RK13 and BSC-40 cells were infected with passaged virus populations in duplicate at MOI = 0.01 for 48 hr and viruses were titered on RK13 + E3 + K3 cells in duplicate.

## Screening for viruses that captured mCherry-E3L

RK13-mCherry-E3L cells were infected with VC-R2 (MOI = 0.01) for 3 days. Viruses were titered on RK13 + E3 + K3 cells. Subsequently, viruses were screened on confluent monolayers of RK13 cells in 6-well plates. Cells were infected at MOI = 0.2 to screen for fluorescence to avoid cell death observed at higher MOIs. For most isolates, plaques were identified after 3 days but some took as long as 5 days. Initially, plates were screened for double-positive mCherry and EGFP expressing plaques using an inverted microscope (Leica DMi8 automated, total magnification ×6.25) using the FITC filter (460–500 nm) to visualize EGFP and the Texas Red filter (540–580 nm) to visualize mCherry. In order to increase throughput, screening for double-positive plaques was then performed with an automated microscope (EVOS FL Auto 2, Thermo Fisher Scientific) using comparable filters. Double-positive virus plaques were picked using a sterile pipette tip. Harvested plaques were subjected to three rounds of

freeze–thaw followed by sonication (two times 30 s). These viruses were then serially diluted to infect RK13 cells for a total of three rounds of plaque purification. After the third round of plaque purification, viruses were amplified in RK13 cells and titered on RK13 + E3 + K3 cells. Viral genomic DNA was isolated from infected RK13 cells according to a published protocol (*Seto et al., 1987*).

## Detection of mCherry-E3L integration sites in the virus genome

The mCherry-E3L integration sites in the purified HGT virus genomes were located by inverse PCR amplification and by PacBio sequencing. For inverse PCR, we used a *Xba*I restriction site in between mCherry-E3L. VC-2 contains 123 *Xba*I sites. Viral DNA from HGT isolates was digested with *Xba*I, and then ligated by T4 DNA ligase (NEB) to make circular DNA. After ligation, E3L-specific, outward-facing primers: HGT-E3L-1F (5′-TGG CAG TAG ATA AAC TTC TTG GTT ACG-3′) and HGT-E3L-1R (5′-AGC CTC ACA CAC AAT CTC TGC G-3′) were used to amplify DNA circles containing mCherry-E3L and neighboring genomic viral DNA. These amplicons were gel purified and Sanger sequenced to define the approximate mCherry-E3L integration sites, because the long poly(A) tails precluded identification of the virus sequences 3′ of the poly(A) tails. To define integration sites and TSDs, primers (*Supplementary file 2*) from the VACV genome that are adjacent to the integration sites were designed to amplify whole integrated captured sequences, which were then Sanger sequenced. Because for HGT5 and HGT19, PCR amplification of the whole insert did not work, two separate PCR amplification with primers located in E3L and the adjacent vaccinia virus genes were performed and sequenced.

To determine if mCherry-E3L integrated into both sites of the ITR in the C19L/B25R locus in HGT15, PCR with DNA of HGT15 and VC-R2, as control, was performed with primers that combined primer JR179-C19L-2F (5′-CGA GGA CTA TGT TTG GTA TAC TG-3′) located in the ITR, toward the termini of the integration site and primers JR180-C13L-2R (5′-CTG GAC TAT CCA CAC CTG-3′) for the 'left' region of the VACV genome, and JR181-B19R-1F (5′-CTA GTA GAG GCG GTA TCA C-3′), for the 'right' region of the VACV genome outside of the ITR. Both primer combinations showed larger bands when DNA HGT15 DNA was used as template, in comparison to VC-R2 DNA, which is indicative of mCherry-E3L in both ITR copies (*Figure 1—figure supplement 4*).

## Measurement of plaque sizes

Confluent 6-well plates of either RK13 or BSC-40 cells were infected with 100 PFU of either each HGT isolate, VC-R2, or vP872 in triplicate. After 1 hr, cells were overlaid with CMC and incubated at 37°C in 5% $CO_2$ for 3 days followed by crystal violet staining. The stained plates and an adjacent ruler were imaged with the EVOS FL 2 microscope (Thermo Fisher Scientific) using a bright-field filter. Plaque sizes were calculated using Adobe Photoshop CC 2019. To calculate this, a global scale was set by measuring the average number pixels in a 1-mm distance from the imaged rulers (133 pixels = 1 mm, SD = 0.98 pixels) using the Adobe Photoshop CC 2019 ruler tool. The Photoshop quick selection tool was used to mark the perimeter of individual plaques followed by area measurement from the analysis tab. Between 30 and 387 plaques were measured for each virus. Kruskal–Wallis one-way analysis of variance followed by Dunn's test of multiple comparisons was used to compare the mean plaque area of HGT samples with vP872 (alpha = 0.05) by GraphPad Prism (version 8.3.0). Graphs are presented as mean ± standard deviation and adjusted p values are shown.

## Identifying virus complementation

Confluent 6-well plates of RK13 + E3 + K3 cells were infected with 100 PFU of HGT1 through HGT20, and the passaged HGT3 from passages 2, 9, and 17. The percentage of mCherry and EGFP double-positive plaques were determined 3 days after infection using the automated EVOS FL Auto 2 microscope (Thermo Fisher Scientific). Images were taken at 4 × 0.13 NA magnification by using filter channels Texas red (540–580 nm) for mCherry and EGFP (465–495 nm) to visualize EGFP. Single-channel autofocus was used with phase-contrast using the first field to autofocus all other fields. 100% of well areas were photographed, stitched and tiled by using the EVOS FL Auto 2 software and analyzed to quantify the number of single-positive EGFP plaques and double-positive EGFP and mCherry expressing plaques. For each virus sample more than 86 plaques were analyzed.

For the VC-R2 and HGT13 complementation assay, confluent 6-well plates of RK13 cells were coinfected in duplicate with each virus (MOI = 0.05 for each virus), or with HGT13 alone (MOI = 0.1). Images for mCherry fluorescence were taken 3 days after infection with EVOS FL Auto 2.

## Detection of tandem duplications of the H4L locus using PCR

Genomic DNA from HGT3 passages 0, 2, 9, and 17 as well as VC-R2 were used as templates for PCR using outward-facing primers: JR81-H4L-2F (5′-GTC TAG TAG ATA TGC TTT TAT TTT TG-3′) and JR80-H4L-2R (5′-CGA AAA TAT AAC TCG TAT TAA AGA G-3′) or JR42-H4L-3F (5′-CAC GGA GAT GGC GTA TTT AAG AG-3′) and JR88-H4L-3R (5′-GAG CTA ACG TGT GAC GAA G-3′). PCR amplified products were cloned into the pCR2.1-TOPO-TA (Invitrogen) vector for amplification and Sanger sequencing using M13F (-21) and M13R primers (UC Davis sequencing facility).

## Library preparation, PacBio CCS sequencing, and data analysis

HiFi SMRTbell library construction and sequencing were performed using Sequel II System 2.0 with P2/C2 (polymerase 2.0 and chemistry 2.0) chemistry at UC Davis DNA Technologies Core according to the manufacturer's instructions (Pacific Biosciences). Before library preparation, the quality of genomic DNA was evaluated by pulsed-field gel electrophoresis (Sage Science Pippin Pulse) and genomic DNA concentration was quantified using a Qubit 3.0 Fluorometer (Life Technologies Q33216). Briefly, 10 µg of each viral genomic DNA was used for HiFi SMRTbell library preparation by using SMRTbell Express Template Prep Kit 2.0 (Pacific Biosciences 100-938-900). The genomic DNA was sheared to ~15–20 kb size by Megaruptor (Diagenode B06010001). The sheared DNA samples were then concentrated and purified by AMPure PB Beads (Pacific Biosciences 100-265-900) by adding 0.45× volume of AMPure PB magnetic beads to each sheared DNA samples and subsequent 80% ethanol precipitation and elution with 100 µl elution buffer. Following genomic DNA shearing and purification, each genomic DNA sample was subjected to single-strand DNA (ssDNA) overhang removal and DNA damage repair. In short, ssDNA overhangs were removed using ssDNA overhang removal reaction mix (DNA prep buffer, NAD, DNA prep additive and DNA prep enzyme) and incubated at 37°C for 15 min. The DNA fragments of each sample were then repaired by using a DNA damage repair mix and incubated at 37°C for 30 min before proceeding to End repair/A-tailing by mixing with 1× End prep mix and incubated at 20°C for 10 min followed by 30 min incubation at 65°C. After that, different barcoded adapters were ligated to each fragmented viral DNA sample by mixing with adapter ligation reaction mix (Overhang adapter v3, ligation mix, ligation additive, and ligation enhancer) and incubated at 20°C for an hour followed by incubation at 65°C for 10 min to inactivate the ligase. After ligation reaction, each adapter-ligated SMRTbell library was treated with exonucleases to remove damaged or unligated DNA fragments for 1 hr at 37°C followed by a second purification step with 0.45× AMPure PB Beads. After purification, the libraries were multiplexed into two library pools for size selection: 6-plex pool 1 and 10-plex pool 2. These pooled SMRTbell libraries were then purified by mixing with 0.45× volume of AMPure PB beads followed by 80% ethanol wash and elution with 31 µl EB for size selection. Size selection was done using the BluePippin Size-Selection System (Sage Science BLU0001) to remove <6 kb SMRTbell templates and eluted the size selected libraries into two size fractions: 9–13 and >15 kb per pooled library. The only fraction with >15 kb SMRTbell libraries was purified with 0.50× AMPure PB Beads and used for primer annealing and polymerase binding. Sequencing Primer v2 (PN 101-847-900) was annealed to each size selected SMRTbell library fraction of >15 kb length at primer: template ratio of 20:1 by denaturation step at 80°C for 150 min followed by slow cooling at 0.1°C/s to 25°C. Prior to sequencing, the polymerase-template complex was bound to the P2 enzyme with 10:1 polymerase to the SMRTbell template ratio for 4 hr. The polymerase-bound SMRTbell libraries were then sequenced by placing libraries in an 8 M SMRT cell at a sequencing concentration of 63 pM and 30 hr movie run time in a Sequel II System 2.0 machine.

For data analysis, PacBio CCS reads were first demultiplexed using lima from SMRT Link 8.0 command-line tools. For all samples, the location of gene H4L and mCherry-E3L sequences were identified in each CCS reads by using BLAST (v2.9+) (Camacho et al., 2009). The search results from BLAST were processed using custom scripts to identify the different modes of mCherry-E3L insertions, as well as the alterations of gene structures of H4L. Representative sequences of CCS reads were extracted to showcase selected scenarios. Those sequences were further analyzed using Seqbuilder, Seqman Pro, Editseq of DNASTAR software package (DNASTAR, Madison, WI). A logo plot from 19 TSDs (first 6 nucleotides 5′ and 3′ of the TSD) and 10 nucleotides of adjacent sequences was created with WebLogo (Crooks et al., 2004). PacBio CCS reads were submitted to ArrayExpress (accession: E-MTAB-9682).

## Acknowledgements

This work was supported by grant AI146915 (to SR) from the National Institute of Allergy and Infectious Diseases, National Institutes of Health. We thank Dr. Adam Geballe, Dr. Bertram Jacobs, and Dr. Bernard Moss for cell lines and viruses, and current and former Rothenburg lab members for helpful discussions.

## Additional information

### Funding

| Funder | Grant reference number | Author |
| --- | --- | --- |
| National Institute of Allergy and Infectious Diseases | AI146915 | Stefan Rothenburg |

The funders had no role in study design, data collection, and interpretation, or the decision to submit the work for publication.

### Author contributions

M Julhasur Rahman, Data curation, Formal analysis, Validation, Investigation, Methodology, Writing - original draft, Writing - review and editing; Sherry L Haller, Investigation, Methodology; Ana MM Stoian, Data curation, Formal analysis, Validation, Investigation; Jie Li, Conceptualization, Data curation, Formal analysis, Supervision, Methodology, Writing - original draft, Project administration, Writing - review and editing; Greg Brennan, Stefan Rothenburg, Conceptualization, Data curation, Supervision, Funding acquisition, Validation, Methodology, Writing - original draft, Project administration, Writing - review and editing

### Author ORCIDs

Greg Brennan (ID) http://orcid.org/0000-0002-4339-9045
Stefan Rothenburg (ID) http://orcid.org/0000-0002-2525-8230

### Decision letter and Author response

Decision letter https://doi.org/10.7554/eLife.63327.sa1
Author response https://doi.org/10.7554/eLife.63327.sa2

## Additional files

### Supplementary files

• Supplementary file 1. Target site duplications identified in horizontal gene transfer (HGT) viruses.

• Supplementary file 2. Oligonucleotides used to amplify the integration sites of mCherry-E3L in horizontal gene transfer (HGT) viruses.

• Transparent reporting form

### Data availability

Sequencing data have been deposited in ArrayExpress under accession code E-MTAB-9682.

The following previously published dataset was used:

| Author(s) | Year | Dataset title | Dataset URL | Database and Identifier |
| --- | --- | --- | --- | --- |
| J Rahman M, Haller SL, Li J, Brennan G, Rothenburg S | 2020 | Cascade evolution in poxviruses: retrotransposon-mediated host gene capture, complementation and recombination | https://www.ebi.ac.uk/arrayexpress/experiments/E-MTAB-9682 | ArrayExpress, E-MTAB-9682 |

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
