## [Editor Report]

This landmark paper reports real-time gene acquisition by vaccinia virus, a DNA virus that replicates in the cytoplasm of infected host, from the host DNA genome. The compelling evidence comes from the rescue of a defective vaccinia virus with a cell line that provides an essential function. Then, horizontal gene transfer that bears sequence hallmarks of LINE-1 transposition and subsequent recombination with sibling genomes are required to generate viable genomes. Detection and description of these combinations of rare events is a technical feat that will be of great interest to anyone interested in human or viral evolution.

---

## [Decision Letter]

**Decision letter after peer review:**

Thank you for submitting your article "Cascade evolution in poxviruses: retrotransposon-mediated host gene capture, complementation and recombination" for consideration by *eLife*. Your article has been reviewed by 2 peer reviewers, and the evaluation has been overseen by a Reviewing Editor and Patricia Wittkopp as the Senior Editor. The following individuals involved in review of your submission have agreed to reveal their identity: Eugene V. Koonin (Reviewer #2); Derek Walsh (Reviewer #3).

The reviewers have discussed the reviews with one another and the Reviewing Editor has drafted this decision to help you prepare a revised submission.

Summary:

Rothenburg and colleagues report acquisition of genes from the host by vaccinia virus via a LINE-2-dependent reverse transcription mechanism. They show that the vaccinia virus-encoded PKR inhibitor E3L that is integrated into the host genome as a fusion with a reporter can be reacquired by a PKR-sensitive virus mutant with the frequency of about 1 in 20 million. Obviously, detection of such rare events is a major experimental achievement.

This novel experimental system to study host gene acquisition by poxviruses will clearly be of broad relevance and utility to other viral systems in the future. Using the system, the authors elegantly illustrate RNA-mediated gene transfer through LINE-1, demonstrating that it not only occurs relatively randomly but also occurs in essential genes. When this occurs, initial propagation of the mutant is facilitated by co-infection with a parental virus until a secondary recombination event between the parental and mutant virus drives the formation of a replication competent virus with the new host gene.

The manuscript is well written, the experimental system is solid, the experimental approach is well controlled and the results are exciting. The comments below are simply requests for further exposition.

1. One of the reviewers has an issue with the interpretation of 'cascade evolution' occurring in the results reported here. Here is the comment; 'The recipient viruses with defects in the viruses with insertions into essential genes are nonviable but they can be rescued by complementation with a virus containing the respective intact gene. Such complementation is part of a process that the authors denote 'cascade evolution' whereby complementation is followed by recombination reintroducing an intact copy of the essential gene into the virus genome containing the insert." This objection seems to rest on two necessarily artificial aspects of the experimental set-up:

1) The extremely strong selection pressure that necessitates complete complementation, and

2) the possibility that the scar left in the defective genome provides a very favorable site, complete with promoter, for homologous insertion. These arrangements were, of course, necessary to observe this process experimentally in real time.

Discussion that might prove helpful to readers might include:

1) The idea that rescue and complementation could occur naturally in both quasispecies generated within a cell and in the en bloc transmission that can occur during cell-to-cell spread;

2) complete failure to grow is clearly a very strong selective pressure that, here, can serve as a stand-in for weaker selective pressure that may not require the presence of complementing or rescuing genomes;

3) the insertion of the host gene into this 'favorable' site merely serves to increase the frequency of the event, because one can easily envisage favorable sites occurring naturally, although clearly at lower frequency.

These points are made here and there in the manuscript but perhaps could be made more explicitly in a discussion of frequency in the Discussion.

2. There was some discussion of whether the RT steps in LINE-1 processes occur in the nucleus or the cytoplasm, and that some poxviruses carry remnants of retroviral genomes can support either argument. For non-specialists, it might be helpful to discuss this a little to fully understand the process.

3. Reviewers request further introduction and citation of the literature on HGT in the Introduction – perhaps this could assist with point (2) as well.

4. It was pointed out that 'Poxviridae' is a family of viruses, not a genus.

*Reviewer #1 (Recommendations for the authors):*

Rothenburg and colleagues report acquisition of genes from the host by vaccinia virus via a LINE-2-dependent reverse transcription mechanism. They show that the vaccinia virus-encoded PKR inhibitor E3L that is integrated into the host genome as a fusion with a reporter can be reacquired by a PKR-sensitive virus mutant with the frequency of about 1 in 20 million. Obviously, detection of such rare events is a major experimental achievement. They further demonstrate that E3L is inserted into the virus genome more or less uniformly including into the essential genes located in the central part of the genome.

Obviously, viruses with insertions into essential genes are nonviable but they can be rescued by complementation with a virus containing the respective intact gene. Such complementation is part of a process that the authors denote 'cascade evolution' whereby complementation is followed by recombination reintroducing an intact cope of the essential gene into the virus genome containing the insert. This is where I have some difficulty accepting the results of this work as presented. 'Cascade evolution' implies an evolutionary mechanism that is important in nature. I doubt that this is the case here. After all, the consolidation of the essential genes in the central portion of the genome is a common phenomenon in NCLDV, with virtually no exceptions. Therefore, I believe that, however interesting, 'cascade evolution' is unlikely to be an important mechanism in nature, and therefore, may not merit being highlighted in the title of this paper and presented as a major finding unlike the mechanism of gene capture itself which undoubtedly is an important even if predictable discovery.

Poxviridae is a family of viruses not a genus as indicated in the Introduction.

Citation of the literature on HGT in the Introduction in inadequate.

*Reviewer #2 (Recommendations for the authors):*

In this manuscript, Rahman and colleagues describe an experimental system to study host gene acquisition by poxviruses, which will clearly be of broader relevance and utility to other viral systems in the future. Using the system, the authors elegantly illustrate RNA-mediated gene transfer through LINE-1, demonstrating that it not only occurs relatively randomly but also occurs in essential genes. When this occurs, initial propagation of the mutant is facilitated by co-infection with a parental virus until a secondary recombination event between the parental and mutant virus drives the formation of a replication competent virus with the new host gene. This is paradigm shifting, as it is widely assumed HGT into an essential gene is lethal and simply doesn't happen. The manuscript is well written, the experimental system is solid and the experimental approach is well controlled. Indeed, the approach is relatively straightforward and as such, I have no experimental concerns. I would suggest that a few points might be considered as additions to the discussion for a general audience.

Many people consider RT steps in LINE-1 processes to occur in the nucleus. The fact that RNA-mediated HGT occurs for poxviruses that replicate in the cytoplasm suggests that the PIC, with RNA/Integrase/RT likely also function in the cytoplasm and can be picked up there? Indeed, some poxviruses carry remnants of retroviral genomes, supporting such events naturally occurring. For non-specialists, it might be helpful to discuss this a little to fully understand the process.

A gene transfer frequency of 1 in 23 million is mentioned. I think it would be worth mentioning that this might be on the higher end as this is an experimental system that is based on strong selective pressure (which will arise in some natural contexts) and providing viral promoter elements etc. There is absolutely nothing wrong with this experimentally, but discussing this might provide some broader context to readers and ultimately, even a far lower frequency is meaningful in terms of the evolution and adaptation of viruses – it only takes one event to be significant.

Could the different behaviors in BSC40's relate to the fact that the original evolution occurred in RK13 cells? Might the host influence the insertion sites that are best tolerated to some extent? As in, perhaps some target sites are actually important in BSC40's but not RK13 cells?

---

## [Author Response]

Reviewer #1 (Recommendations for the authors):Rothenburg and colleagues report acquisition of genes from the host by vaccinia virus via a LINE-2-dependent reverse transcription mechanism. They show that the vaccinia virus-encoded PKR inhibitor E3L that is integrated into the host genome as a fusion with a reporter can be reacquired by a PKR-sensitive virus mutant with the frequency of about 1 in 20 million. Obviously, detection of such rare events is a major experimental achievement. They further demonstrate that E3L is inserted into the virus genome more or less uniformly including into the essential genes located in the central part of the genome.Obviously, viruses with insertions into essential genes are nonviable but they can be rescued by complementation with a virus containing the respective intact gene. Such complementation is part of a process that the authors denote 'cascade evolution' whereby complementation is followed by recombination reintroducing an intact cope of the essential gene into the virus genome containing the insert. This is where I have some difficulty accepting the results of this work as presented. 'Cascade evolution' implies an evolutionary mechanism that is important in nature. I doubt that this is the case here. After all, the consolidation of the essential genes in the central portion of the genome is a common phenomenon in NCLDV, with virtually no exceptions. Therefore, I believe that, however interesting, 'cascade evolution' is unlikely to be an important mechanism in nature, and therefore, may not merit being highlighted in the title of this paper and presented as a major finding unlike the mechanism of gene capture itself which undoubtedly is an important even if predictable discovery.

We agree with the reviewer that the focus on cascade evolution in the title somewhat distracts from the major finding of retrotransposon-mediate gene capture. We therefore changed the title to:

“LINE-1 retrotransposons facilitate horizontal gene transfer into poxviruses”.

Poxviridae is a family of viruses not a genus as indicated in the Introduction.

We have made this change in the text (Line 72).

Citation of the literature on HGT in the Introduction in inadequate.

We included additional examples of the current computational methods used to predict

HGT in the introduction, specifically phylogenetic comparisons and nucleotide composition analysis (Lines 45-59), to complement our discussion of vIL-10 and LINE-1 mediated integration of SINE into taterapox virus.

“So far, the detection of HGT in large DNA viruses has relied on bioinformatic approaches, primarily phylogenetic and sequence composition analyses. For example, Odom, et al., performed a family wide comparison of poxvirus coding genes to different taxonomic subsets, such as “all eukaryotic genes” or “all other viral genes”. They found that poxvirus ORFs are on average more similar to eukaryotic genes than other viruses, suggesting substantial HGT into poxviruses over evolutionary time (Odom et al., 2009). In an orthogonal approach, Monier, et al., used a Bayesian methodology to identify large DNA virus genes with anomalous nucleotide composition relative to the viral genome as a whole.

Using this approach, they determined that a large number of the compositionally anomalous genes in *Poxviridae* are associated with host immune control (Monier et al., 2007). From these and other studies, it has become apparent that viral homologs of host interleukin 10 (IL-10) genes are among the best supported examples of these HGT events. Viral IL-10 genes, which presumably provide a selective advantage through inhibition and modulation of the antiviral response, appear to have been independently acquired by several herpesviruses and poxviruses (Hughes and Friedman, 2005, Schonrich et al., 2017). While these methods can detect putative HGT they can do little to elucidate the frequency or mechanisms of HGT.”

Reviewer #2 (Recommendations for the authors):In this manuscript, Rahman and colleagues describe an experimental system to study host gene acquisition by poxviruses, which will clearly be of broader relevance and utility to other viral systems in the future. Using the system, the authors elegantly illustrate RNA-mediated gene transfer through LINE-1, demonstrating that it not only occurs relatively randomly but also occurs in essential genes. When this occurs, initial propagation of the mutant is facilitated by co-infection with a parental virus until a secondary recombination event between the parental and mutant virus drives the formation of a replication competent virus with the new host gene. This is paradigm shifting, as it is widely assumed HGT into an essential gene is lethal and simply doesn’t happen. The manuscript is well written, the experimental system is solid and the experimental approach is well controlled. Indeed, the approach is relatively straightforward and as such, I have no experimental concerns. I would suggest that a few points might be considered as additions to the discussion for a general audience.Many people consider RT steps in LINE-1 processes to occur in the nucleus. The fact that RNA-mediated HGT occurs for poxviruses that replicate in the cytoplasm suggests that the PIC, with RNA/Integrase/RT likely also function in the cytoplasm and can be picked up there? Indeed, some poxviruses carry remnants of retroviral genomes, supporting such events naturally occurring. For non-specialists, it might be helpful to discuss this a little to fully understand the process.

We added a statement (lines 239-246) stating that:

“LINE-1-mediated reverse transcription is generally thought to occur in the nucleus. Because poxviruses replicate exclusively in the cytoplasm, the detected signatures of LINE-1-mediated retrotransposition after HGT reported here suggests that LINE-1 RT and integrase activities are not limited to the nucleus. The notion that LINE-1s can facilitate the transfer of host genetic material into poxviruses through an RNA-mediated process during the natural course of viral infection is further supported by the presence of a naturally-occurring SINE element in taterapox virus, which is surrounded by a 16 bp target site duplication (Piskurek and Okada, 2007).”

A gene transfer frequency of 1 in 23 million is mentioned. I think it would be worth mentioning that this might be on the higher end as this is an experimental system that is based on strong selective pressure (which will arise in some natural contexts) and providing viral promoter elements etc. There is absolutely nothing wrong with this experimentally, but discussing this might provide some broader context to readers and ultimately, even a far lower frequency is meaningful in terms of the evolution and adaptation of viruses – it only takes one event to be significant.

We agree with the reviewer. To address this, we have added a paragraph to the discussion (Lines 321-330):

“The calculated approximate transfer rate of E3L-mCherry into VACV in this experimental setting of 1 in 23 million viable virions, is probably an underestimation, as integrations in essential loci have detrimental effects on virus replication unless complementing viruses are present. It should be noted that we used a system with a strong synthetic poxvirus promoter and a strong VACV PKR inhibitor in order to detect HGT. Retention of a transferred host gene that has not been optimized is expected to happen with a far lower frequency. However, we selectively looked for the uptake of only one of several thousand genes that are expressed by the host cells. Thus, HGT of host genes, including transposable elements such as SINEs and LINEs, into poxviruses can be expected to occur at a high frequency, but in most cases the inserted gene will not be maintained because of negative or neutral effects on virus replication.”

Could the different behaviors in BSC40’s relate to the fact that the original evolution occurred in RK13 cells? Might the host influence the insertion sites that are best tolerated to some extent? As in, perhaps some target sites are actually important in BSC40’s but not RK13 cells?

It is possible that the insertion site is influenced by the cell type, which will be addressed by future studies. Because complementation of replication-deficient viruses was more efficient in RK13 as opposed to BSC-40 cells, it appears that propagation of viruses with insertions in essential genes would be favored in the former cells. We added a paragraph to the discussion (Lines: 261-274):

“The concentration of insertions around the H4L locus might indicate that some regions of the genome are indeed hotspots for integration. However, the number of integration events detected in this study precludes a definitive answer, and will require further analysis of more HGT events. It is also possible that the insertion site is influenced by the cell type. Because complementation of replication-deficient viruses was more efficient in RK13 as opposed to BSC-40 cells, it appears that propagation of viruses with insertions in essential genes might be favored in the former cells. In a companion paper, Fixsen et al., used a similar strategy to detect HGT by expressing the other VACV PKR inhibitor K3 in RK13 cells, which were also used in our study, and selecting for virus replication in Syrian hamster BHK cells. They found a lower proportion of HGT integrations in the central genome region (Fixsen et al., 2020). The reasons for the different integration patterns could be due to the different cells used for selection and might reflect differences in the relative importance of viral genes in these cells or different, cell line-dependent complementation efficiencies.”

Other points raised by the editors not covered in the above response to reviewers:Complete failure to grow is clearly a very strong selective pressure that, here, can serve as a stand-in for weaker selective pressure that may not require the presence of complementing or rescuing genomes

We have emphasized this point in the Discussion (Lines 277-287) stating that:

“Our experimental system relies on the complete failure for viruses to replicate in the absence of a PKR inhibitor, creating a strong selective pressure*.* In eight instances in this study we also identified integrations into essential genes, creating a similarly strong selective pressure against virus replication. In these cases, complementation followed by recombination was sufficient to overcome this selective pressure in vitro. The homology regions surrounding the cowpox virus GAAP gene (Figure 6) suggest a similar process can occur during natural infection as well. In instances in which the selective pressure to maintain the horizontally transferred gene is weaker or the inactivated gene is non-essential but still contributes to fitness, we hypothesize that similar molecular events might occur. This balance between the negative effects of genome disruption and improved fitness might also display some degree of cell- and species-specificity.”

Please indicate in the Materials and methods section of your manuscript if [your cell line] identity has been authenticated, state the authentication method (such as STR profiling), and report the mycoplasma contamination testing status.

We added the following statements to the methods section:

Cell lines used in this study were negative for mycoplasma contamination as determined with Lookout Mycoplasma PCR Detection Kit (Millipore Σ). During the course of this study, the RK13 cells we used were confirmed to be of European rabbit (*O. cuniculus*) origin by PacBio sequencing (ArrayExpress accession: E-MTAB-9682). PKR expressed in BSC-40 was amplified from cDNA and sequenced, which confirmed that the cells are of African green monkey (*Chlorocebus aethiops*) origin.

We have added this information to the text (Lines 355-360).